# BEYOND LINEARITY IN ATTENTION PROJECTIONS: THE CASE FOR NONLINEAR QUERIES[*]

**Marko Karbevski**
marko.karbevski@gmail.com

## ABSTRACT

Recent algebraic analysis shows that in decoder-only and encoder-only transformers, the Query projection $W_Q$ may be set to identity without noticeable performance deterioration. This is possible because attention depends on $X$ only through the products $XW_Q, XW_K, XW_V$, allowing basis transformations to be absorbed by adjacent layers and propagated through the network. We replace $W_Q \in \mathbb{R}^{d \times d}$ with a nonlinear residual of the form $Q(X) = X + f_\theta(X)$, where $f_\theta$ is a bottleneck MLP with $d^2 + O(d)$ parameters. The identity term anchors the nonlinearity to a known-good prior. Experiments on GPT-3 small style models show consistent improvement over the baseline, comfortably outperforming a model with 12.5% more non-embedding parameters. These results motivate investigation at larger scales and across modalities.

## 1 INTRODUCTION

The transformer architecture (Vaswani et al., 2017) computes attention via Query, Key, and Value projections, each parameterized by learned weight matrices $W_Q, W_K, W_V \in \mathbb{R}^{d \times d}$. Recent work has revealed a fundamental invariance in the attention computation that renders the Query projection algebraically redundant. This observation, grounded in the geometry of the transformer computation graph, motivates a principled modification: replacing the redundant linear projection with a nonlinear one.

**Algebraic and topological structure.** Consider self-attention transformers with arbitrary masking patterns, which subsumes encoder-only architectures, decoder-only architectures, and certain cross-attention configurations such as prefix language models where the prefix serves as encoder context. Let multi-head attention be defined as $\text{MHA}(X) = \text{Concat}(\text{head}_1, \dots, \text{head}_h)W_O$, where each head computes $\text{head}_i = \text{Attention}(XW_Q^i, XW_K^i, XW_V^i)$ with $W_Q^i, W_K^i, W_V^i$ the per-head projection matrices. Writing $W_Q = (W_Q^1 | \cdots | W_Q^h)$ and similarly for $W_K, W_V$, attention depends on $X$ only through the products $XW_Q, XW_K, XW_V$. Karbevski & Mijoski (2025) establish that this admits a reparametrization invariance: for any invertible $\Theta$,

$$(X, W_Q, W_K, W_V, W_O) \mapsto (X\Theta, \Theta^{-1}W_Q, \Theta^{-1}W_K, \Theta^{-1}W_V, W_O) \tag{1}$$

leaves the MHA output unchanged. Since singular matrices have Lebesgue measure zero, $W_Q, W_K,$ and $W_V$ are almost surely invertible, allowing the choice $\Theta = W_Q$ (or $W_K, W_V$) to simplify one of the three matrices. Under mild conditions on the network topology, this basis change propagates through the entire network: each layer passes its transformation to its predecessor, telescoping back to the embedding layer. Graef (2024) first proved this propagation for simplified architectures; Karbevski & Mijoski (2025) extended it to standard transformers with layer normalization and residual connections, verifying $W_Q = I$ empirically. Setting $\Theta = W_Q$ yields $W_Q \mapsto I$, establishing that $W_Q$ is algebraically redundant. Crucially, this extends to Grouped Query Attention: since GQA shares $W_K$ and $W_V$ across query groups, only $W_Q$ can be eliminated without disrupting the shared structure, making queries the natural choice. The invariance is also compatible with Mixture-of-Experts. While we test with learned positional embeddings, this reparametrization also holds for RoPE (Su et al., 2024), as shown by Karbevski & Mijoski (2025).

---

[*]Code and checkpoints: https://github.com/MarkoKarbevski/beyond_query_linearity

**Identity as a good prior.** Empirically, models trained with $W_Q = I$ match baseline performance while remaining stable at $3\times$ lower weight decay (Karbevski & Mijoski, 2025). The identity is thus a good prior for queries, both for representability and for training stability.

**Proposal.** The work of Graef (2024) and Karbevski & Mijoski (2025) establishes that any linear expenditure of parameters at $W_Q$ is largely redundant: the learned matrix can be absorbed into adjacent layers without changing the function computed. If we wish to allocate parameters directly to the query pathway, nonlinearity is therefore imposed. Following the suggestion of Karbevski & Mijoski (2025), we implement a nonlinear query projection directly:

$$Q(X) = (X + f_\theta(X))/2 \tag{2}$$

where $f_\theta$ is a bottleneck MLP with $d^2 + O(d)$ learnable parameters, the same order as the $W_Q$ it replaces. The $1/2$ scaling follows Karbevski & Mijoski (2025). The identity term $X$ anchors the nonlinearity to a known-good prior while providing a direct path for gradient flow (He et al., 2016), preserving the stability established for $W_Q = I$. Keys and values remain standard linear projections. Specifically, we implement $f_\theta$ as:

$$f_\theta(X) = \mathrm{LN}\left(\mathrm{GELU}\left(\mathrm{RMSNorm}(X)W_1\right)W_2\right) \tag{3}$$

where $W_1 \in \mathbb{R}^{d \times r}$, $W_2 \in \mathbb{R}^{r \times d}$, and $r = d/2$. The query $Q(X) = (X + f_\theta(X))/2$ is applied independently per token. The attention logits $Q^i {K^i}^\top / \sqrt{d_k}$ per head use standard scaling. Matrix parameters total $2dr = d^2$; normalization layers add $O(d)$ parameters ($< 0.1\%$ overhead). The design prioritizes stability over throughput or final quality; we conjecture that this formulation may be overly conservative. The choice of LN and RMSNorm was made for stability in initial experiments; systematic exploration of normalization variants, or their elimination entirely, is an immediate priority for future work.

**Related work.** Kernel-based attention methods (Choromanski et al., 2021; Katharopoulos et al., 2020) apply nonlinear feature maps $\phi(Q), \psi(K)$ to achieve linear complexity, but retain standard linear projections $Q = XW_Q$ before the nonlinearity. MLP-Attention (Morsali et al., 2023) eliminates Q and K projections entirely, replacing the $QK^\top$ computation with an MLP that maps embeddings directly to attention weights; however, this adds substantial parameters ($\sim 10\%$), was validated only at small scale ($\sim 5M$ parameters on character-level Tiny Shakespeare), uses no residual structure, and lacks theoretical motivation. Zhang (2023) replace all of Q, K, V with two-layer feedforward MLPs (no residual), also adding substantial parameters; they fine-tune pretrained models rather than training from scratch, and offer no algebraic justification. Nonlinear extensions of LoRA (Li et al., 2024; Ji et al., 2024; Dong et al., 2025) introduce nonlinearities to break the low-rank bottleneck of adapters in parameter-efficient fine-tuning; this is a distinct setting from base architecture design, with different goals and justification. To our knowledge, this is the first direct implementation of nonlinear query projections as suggested by Karbevski & Mijoski (2025): a principled, parameter-neutral architectural modification with residual structure for pretraining, where nonlinearity is introduced because there is strong empirical and theoretical evidence that learned linearity is redundant.

## 2 MOTIVATION

### 2.1 LINEAR PROJECTIONS AND THE DECODING BOTTLENECK

Let $X \in \mathbb{R}^{n \times d}$ denote the input to an attention layer, where $n$ is the sequence length and $d$ is the model dimension. Write $x \in \mathbb{R}^{1 \times d}$ for a single token.

The input to each attention layer is the output of the previous block: $x = x_{\mathrm{prev}} + \mathrm{MLP}(x_{\mathrm{prev}})$. From this single $x$, standard attention generates four vectors in $\mathbb{R}^{1 \times d}$: query $q = xW_Q$, key $k = xW_K$, value $v = xW_V$, and residual $x$. All four are linear functions of $x$. This creates a bottleneck: the four outputs cannot vary independently.

Making the query nonlinear, $q = x + f_\theta(x)$, partially decouples it. The query gains independent variation, leaving more freedom in how $x$ serves keys, values, and the residual. This bottleneck may be more pronounced in Mixture-of-Experts architectures, where experts typically operate with reduced latent dimension; we leave this investigation to future work.

## 2.2 Evidence for Functional Specialization

The projections $W_Q$, $W_K$, $W_V$ are not interchangeable. Within each head, if $W_Q^i = W_K^i$, then $Q^i K^{i^\top} = X W_Q^i W_Q^{i^\top} X^\top$ is symmetric; empirically, trained attention matrices are not symmetric. Hu et al. (2022) report that low-rank adaptation of $W_Q$ and $W_V$ jointly yields superior fine-tuning performance, whereas adapting $W_K$ alone is insufficient. Li et al. (2023) observe two-stage training: $\|W_V\|_F$ grows while $\|W_Q\|_F, \|W_K\|_F \approx 0$ in early training; only later do $W_Q, W_K$ begin learning. These findings motivate giving the query pathway its own dedicated computation.

One might ask whether the same residual structure could apply to K and V. This is non-trivial: symmetric skip structure in both Q and K introduces a parameter-free $XX^T$ term biasing attention toward self-similarity. Moreover, Ji et al. (2025) show that self-attention is uniquely ill-conditioned and dependent on skip connections for regularization; removing the skip around attention, as explored by He et al. (2023), might help but requires careful initialization. Asymmetric designs using Hyper-Connections (Zhu et al., 2025; Xie et al., 2025) may resolve this tension. We leave extensions to K and V as future work.

## 3 Experiments

### 3.1 Setup

We train on OpenWebText on a single NVIDIA RTX 5090 GPU. The baseline is NanoGPT (Karpathy, 2023): 12 layers, 12 heads, $d_{\text{model}} = 768$, MLP hidden dimension $3072 = 4d_{\text{model}}$, context length 1024, no biases, tied embedding/LM-head, GPT-2 tokenizer. We write $d = d_{\text{model}}$ throughout. We use AdamW with $\beta_1 = 0.9$, $\beta_2 = 0.95$, 2k steps of warmup, cosine decay, and gradient clipping at 1.0; ~490k tokens per gradient step. Data splits use NanoGPT's provided seeds. To ensure fair comparison, we pre-generate all batch indices from a fixed random seed; every model sees identical training data at each step and identical evaluation data at each validation step. Validation loss is estimated every 1000 steps by averaging over 2400 sequences ($\approx$2.5M tokens).

Training runs for 60k steps ($\sim$29B tokens), far beyond Chinchilla-optimal for 124M parameters ($\sim$2.5B tokens). This regime tests whether improvements persist under extended training, representing "hard" wins rather than gains that might vanish with proper token scaling.

We compare **Residual GELU** ($Q = (X + f_\theta(X))/2$) against the **Baseline** and MLP-widened controls (MLP hidden dimension $4.75d$ instead of $4d$, adding 12.5% non-embedding parameters). This direct comparison isolates architectural benefits from capacity gains, avoiding reliance on scaling law extrapolations (Kaplan et al., 2020; Hoffmann et al., 2022). The baseline uses weight decay 0.1 and learning rate $6 \times 10^{-4}$ decaying to $6 \times 10^{-5}$. Given the stability observed with $W_Q = I$ (Karbevski & Mijoski, 2025), we explored reduced weight decay ($2^{-5} \approx 0.03$) and higher learning rates (up to $3 \times 10^{-3}$) for the nonlinear variant. Full configurations are reported in Table 1.

### 3.2 Results

Figure 1 shows training dynamics from step 1k to 59k. The nonlinear query achieves consistently lower validation loss than baseline and MLP-widened controls. Reduced weight decay ($2^{-5}$) with increased learning rate (up to $3 \times 10^{-3}$) yields the best performance, consistent with implicit regularization from the identity anchor (Karbevski & Mijoski, 2025). At 59k steps, validation loss is 2.915 versus 2.956 for baseline (1.40% relative improvement), comfortably outperforming $\text{MLP}_{4.75}$ (0.98%) which has 12.5% more non-embedding parameters. Curiously, relative gains are largest during warmup, diminish mid-training, then reappear for the best variant in the final phase (Appendix A).

Our model trains stably at weight decay $2^{-5} \approx 0.03$, whereas the baseline diverges before 20k steps at weight decay 0.05 (data available in the code repository). This confirms findings of Andriushchenko et al. (2024) on weight decay necessity; their NanoGPT experiments tested only 0.1 and 0, and we fill the gap by showing instability emerges between these values, in a regime with more tokens per gradient step that should favor stability. The tolerance of higher learning rates may relate to scale-invariance from normalization layers (Lyle et al., 2024).

Table 1: Model configurations and validation loss at 59k steps. All models are based on NanoGPT ($d_{\mathrm{model}} = 768$, $L = 12$, $T = 1024$) without bias. $\mathrm{MLP}_w$ denotes baseline with MLP hidden dimension $w \cdot d$.

| | Baseline | $\mathrm{MLP}_{4.75}$ | $\mathrm{MLP}_{4.75}$ (scaled) | Res. GELU | Res. GELU |
|---|---|---|---|---|---|
| Query | Linear | Linear | Linear | Nonlinear | Nonlinear |
| Non-emb params | 85M | 96M (+12.5%) | 96M (+12.5%) | 85M | 85M |
| Max LR | $6 \times 10^{-4}$ | $6 \times 10^{-4}$ | $5.66 \times 10^{-4}$ | $2.6 \times 10^{-3}$ | $3 \times 10^{-3}$ |
| Min LR | $6 \times 10^{-5}$ | $6 \times 10^{-5}$ | $5.66 \times 10^{-5}$ | $3 \times 10^{-5}$ | $3 \times 10^{-5}$ |
| Weight decay | 0.1 | 0.1 | 0.1 | $2^{-5}$ | $2^{-5}$ |
| Val loss (59k) | 2.956 | 2.927 | 2.928 | 2.919 | **2.915** |
| Rel. improvement | 0 | 0.98% | 0.94% | 1.24% | **1.40%** |

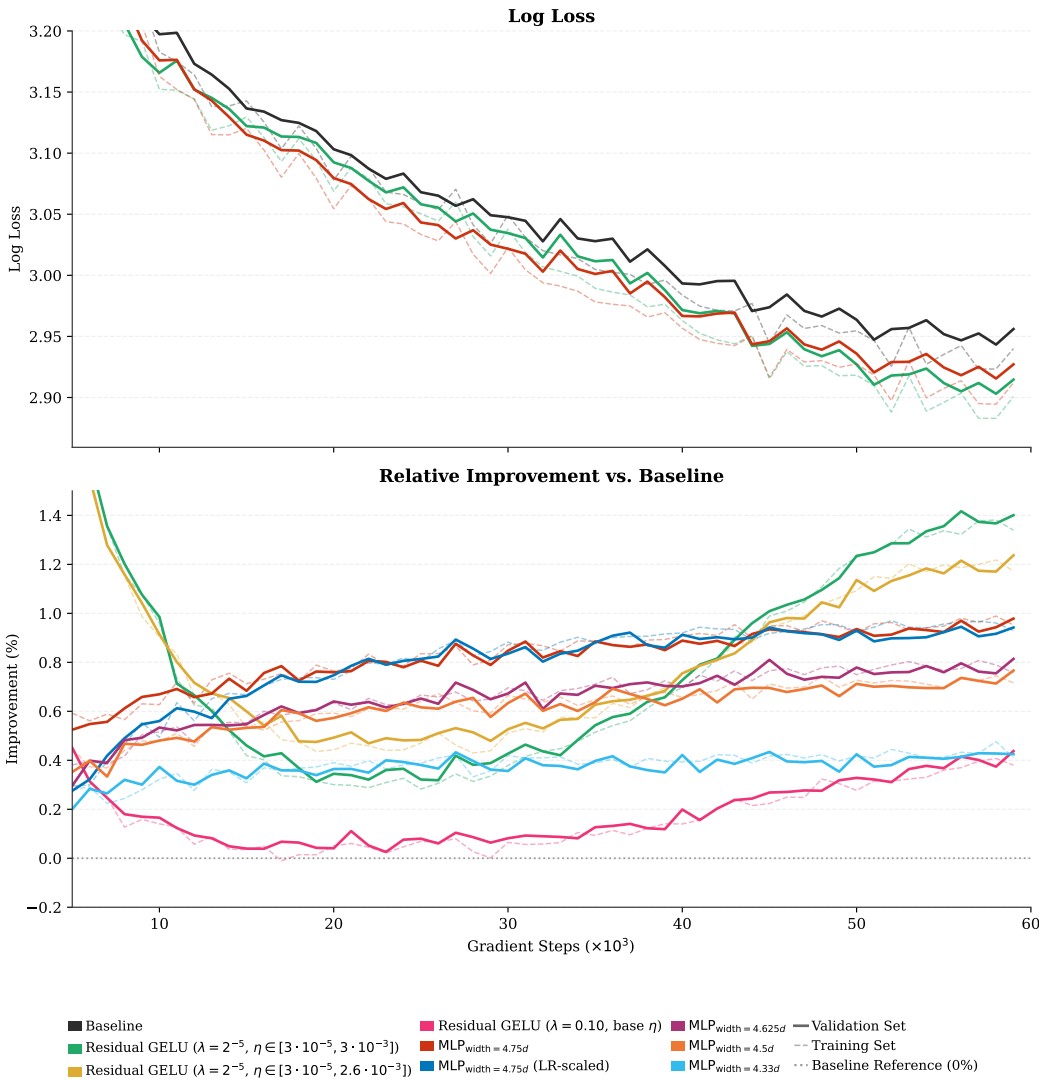

Figure 1: Training dynamics (steps 1k-59k). Solid: validation, dashed: training. *Top:* Loss curves. *Bottom:* Relative improvement over baseline.

## 4  DISCUSSION

**Limitations.** Our experiments use a single model scale ($\sim$124M parameters) with 60k training steps and no systematic hyperparameter search. The hyperparameter landscape (learning rate schedules,

weight decay, bottleneck architecture, LayerNorm epsilon) remains largely unexplored, suggesting that the reported 1.40% improvement represents a floor rather than a ceiling. Inference speed has not been measured. Multiple seeds have not been run; we mitigate this by training and evaluating all models on identical batches, and by training far beyond Chinchilla-optimal ($\sim$29B tokens versus $\sim$2.5B optimal) to reduce the influence of early stochasticity. Larger scales, weight dynamics analysis, and downstream evaluation remain for future work.

**Efficiency and deployment.** The bottleneck MLP replaces $W_Q$ at the same parameter count and FLOPs; overhead from the GELU nonlinearity and normalization layers is negligible. However, the nonlinear query introduces a serial dependency: the bottleneck MLP must complete before attention can proceed. Custom kernel implementations may be needed for practical deployment. Further efficiency is possible: since the first bottleneck layer maps $d \rightarrow r = d/2$, it can be fused with the Key and Value projections, reducing the combined $W_{QKV}$ output from $d \rightarrow 3d$ to $d \rightarrow 2.5d$. ReLU or ReLU$^2$ activations should be tested as alternatives to GELU for additional speed gains.

**Scaling.** As $d$ increases, the bottleneck dimension $r = d/2$ grows proportionally, increasing the expressivity of $f_\theta$. Unlike Primer (So et al., 2021), which achieved improvements through depthwise convolutions with explicit locality bias (a bias larger models learn naturally, making the improvement redundant at scale), there is no obvious path by which scaling would make our approach redundant: the identity anchor provides a natural skip connection, and the bottleneck grows with model size.

**Extensions.** Our results suggest that Karbevski & Mijoski (2025) may benefit from substantially higher learning rates than they considered; their experiments used baseline learning rates, whereas we find the nonlinear architecture tolerates $5\times$ higher values. Systematic comparison with output-side gating mechanisms (Qiu et al., 2025) is another direction. The identity anchor makes our approach compatible with RoPE, MoE, and GQA/MQA (Karbevski & Mijoski, 2025; Graef, 2024). Rather than a separate $f_\theta$, one could derive queries from intermediate MLP activations ($f_\theta(X) = \text{MLP}_{\text{hidden}}(X)_{[0:d]}W'$), reusing computation. For pretrained models, one could anchor to existing projections: $Q = (XW_Q + f_\theta(X))/2$; this relates to nonlinear LoRA extensions (Li et al., 2024; Ji et al., 2024; Dong et al., 2025).

## 5 CONCLUSION

We presented the first direct implementation of nonlinear query projections as suggested by Karbevski & Mijoski (2025): a residual form $Q(X) = (X + f_\theta(X))/2$ that anchors nonlinearity to the identity prior. This design is motivated by the observation that linear $W_Q$ is redundant under mild conditions, yet queries may benefit from dedicated nonlinear computation decoupled from the value stream. At the same parameter budget, the nonlinear query achieves 1.40% lower validation loss than baseline, comfortably outperforming a model with 12.5% more non-embedding parameters. The modification also improves training stability, tolerating weight decay and learning rate settings where the baseline diverges. Our formulation prioritizes stability over throughput or final quality, suggesting these results represent a floor rather than a ceiling. Exploring less conservative designs remains an immediate direction for future work.

### ACKNOWLEDGEMENTS

I am grateful to the anonymous reviewers for their constructive feedback, which improved the clarity of this work. I also thank Nils Graef, Yiping Ji, Haris Mandal, and Antonij Mijoski for valuable discussions.

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

## A  HYPERPARAMETER SWEEP

This section is illustrative and does not claim completeness. We include it to expose the sweep conducted and to highlight the unexpectedly high initial gains across configurations, which stabilize mid-training (Figure 2, steps 1k-59k).

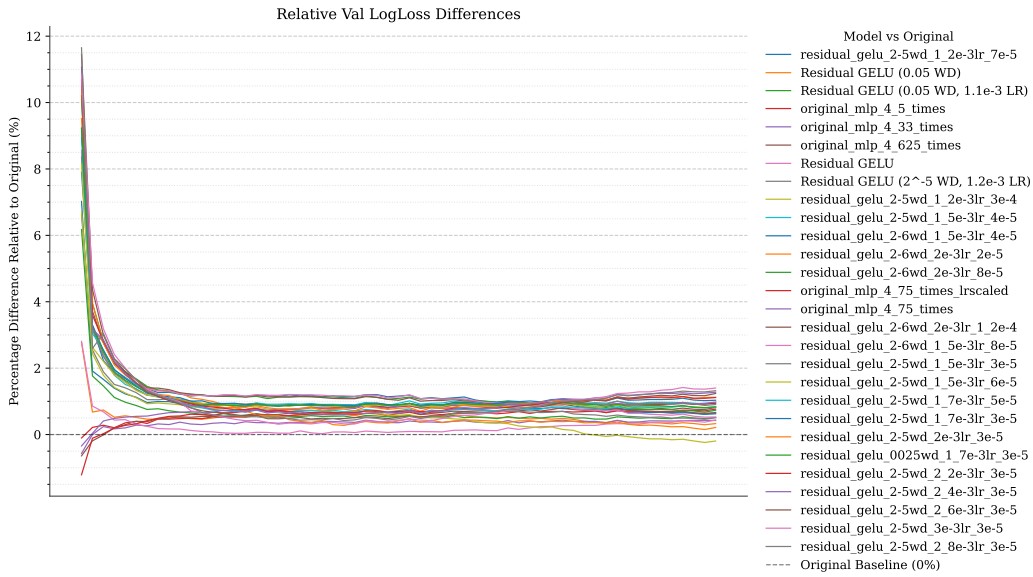

Figure 2: Relative improvement over baseline (steps 1k to 59k). Nonlinear configurations: 84.97M parameters; MLP-widened controls: 89.6 to 95.6M.

