# OpenReview forum: "Beyond Linearity in Attention Projections: The Case for Nonlinear Queries"
_ICLR.cc/2026/Workshop/GRaM — ICLR 2026 Workshop GRaM Poster_

### Official Review · Reviewer_LFED · 2026-02-08

**Rating:** 6
**Confidence:** 3

**Review:**

__Overview__
The paper proposes replacing the standard linear Query projection ($W_Q$) with a nonlinear residual MLP of equivalent parameter count ($d^2$). Motivated by recent algebraic findings that linear $W_Q$ is redundant, the authors utilize this parameter budget to break the linear dependency between Q, K, and V. Experiments on NanoGPT (124M) show consistent improvements in validation loss and training stability.

__Strengths__
- Principled Motivation: The method is grounded in the algebraic redundancy of linear projections rather than arbitrary complexity.
- Parameter Efficiency: It introduces nonlinearity without increasing the model size, strictly adhering to the $O(d^2)$ budget of the matrix it replaces.
- Training Stability: The residual formulation anchored on identity ($Q = (X + f(X))/2$) enables stable training at lower weight decay.

__Weaknesses / Questions__
- Inference Latency: Replacing a single matrix multiplication with a multi-step MLP (LN, GELU, projections) likely increases wall-clock time, which is not discussed.
- Hyperparameter Sensitivity: The method requires specific tuning (e.g., lower weight decay) to outperform the baseline, raising questions about robustness.

__Relevance to topics listed in GRaM call for papers__
Yes

__Originality and novelty__
Repurposing the "redundant" linear query parameters to introduce nonlinearity is a clever and novel geometric insight, distinct from typical kernel-based attention methods.

__Technical soundness__
The theoretical premise is well-cited, and the experimental comparison (matching parameter counts) is fair. The limitations are acceptable for a Tiny Paper track.

__Clarity in writing and organization__
The paper is concise, the mathematical formulation is clear, and it respects the page limit.

__For the Proceedings track__
N/A

__Double-blind reviewing__
No violations of anonymity were found.

__Use of LLMs__
The text is technical and precise; there are no signs of excessive or improper LLM generation.

**Pmlr Suitability:**

NA

---

### Official Review · Reviewer_JVhV · 2026-02-10

**Rating:** 6
**Confidence:** 4

**Review:**

**Overview**
This paper observes that $W_q$ and $W_k$ can be fused together, allowing $W_q$ to be set as the identity matrix. Prior experiments have shown that this can serve as a useful prior. The authors explore a non-linear function to produce the query, specifically $Q(X) = (X + f_\theta(X)) / 2$. Early experiments with NanoGPT show improvements over the baseline.

**Strengths**
1. Strong motivation: introducing non-linearity is well justified by redundancy.
2. The model’s parameter count remains approximately the same before and after the modification.

**Weaknesses**
1. Modern transformers typically use multi-head attention $MHA$, where $W_q$ is not a square matrix. This weakens the argument for setting $W_q$ as the identity, since fusing $W_q$ with $W_k$ could change the shape of $W_k$. Consequently, the roles and training dynamics of $W_q$ and $W_k$ differ compared to the Vallania attention block, where they are square matrices. While this does not affect the non-linear query implementation, it does impact the redundancy-based motivation or equation 1. Adding an analysis on this would strengthen the paper.
2. Most modern transformers employ RoPE between the query and key. Future work could include experiments incorporating RoPE.
3. Inference speed: adding LN and GELU introduces overhead, particularly when applied to every layer. An analysis of throughput, time-to-first-token (TTFT), and FLOPs would be helpful. Preliminary tests could also explore replacing LN and GELU with simpler alternatives to reduce overhead.

**Relevance to topics listed in GRaM call for papers:** Yes

**Originality and novelty:** Yes

**Technical soundness of method:** Yes

**Clarity in writing and organization of the paper** The paper is concise, the mathematical formulation is clear, and it respects the page limit.

**For the Proceedings track:** N/A

**Double-blind reviewing:** No violations of anonymity were found.

**Use of LLMs:** The text is technical and precise; there are no signs of excessive or improper LLM generation.

**Pmlr Suitability:**

NA

---

### Official Review · Reviewer_RFns · 2026-02-24

**Rating:** 3
**Confidence:** 3

**Review:**

**Overview**: The paper proposes adding non-linear residuals to queries in attention projections. The method shows reduced validation loss from the linear baseline.

**Strengths**:
1. Theoretically grounded motivation: The paper chooses non-linearity building on recent algebraic findings.
2. Sound analysis on different attention matrix: The paper provides analysis on the impact of different forms of Q, K, V matrices on training behavior.

**Weaknesses**:
1. Potential mismatch to the workshop's core focus: The paper proposes the non-linear modification to the queries and merely justifies the remmoval of the redundant linear layer. However, this remains in the Euclidean space and does not introduce any geometrically grounded representation or parameters. It's uncertain that whether this paper closely fits the theme of this workshop.
2. Empirical results: Empirical results could be enhanced by larger-scale and longer-step training. The primary and only evaluation metric used is validation loss, which may not be robust for practical use. And the reported improvement seems marginal under the current experimental setup. More setups should be experimented to further validate the result.
3. The related work section is concise, but the paper would benefit from clearer delineation of what is fundamentally new compared to other “nonlinear attention” or “MLP attention” approaches.

**Pmlr Suitability:**

NA

---

### Meta-Review · Area_Chair_DjxL · 2026-02-26

**Decision:**

Accept

**Metareview:**

The paper proposes adding non-linear residuals to queries in attention projections. The method shows reduced validation loss from the linear baseline. Reviewers appreciate the strong motivation, sound analysis, and parameter efficiency of the method. I encourage the authors to incorporate reviewers’ feedbacks including experiment setups, using modern transformer architectures, and measuring inference speed.

**Relevance To Proceedings:**

Tiny paper — does not apply

**Relevance To Workshop:**

Yes — suitable for GRaM

---

### Decision · Program_Chairs · 2026-03-02

Accept (Poster)